# Tlalpan 2020 Case Study: Enhancing Uric Acid Level Prediction with Machine Learning Regression and Cross-Feature Selection

**DOI:** 10.3390/nu17061052

**Published:** 2025-03-17

**Authors:** Guadalupe Gutiérrez-Esparza, Mireya Martínez-García, Manlio F. Márquez-Murillo, Malinalli Brianza-Padilla, Enrique Hernández-Lemus, Luis M. Amezcua-Guerra

**Affiliations:** 1“Researcher for Mexico” Program under SECIHTI, Secretariat of Sciences, Humanities, Technology, and Innovation, Mexico City 08400, Mexico; 2Division of Diagnostic and Treatment Services, National Institute of Cardiology Ignacio Chávez, Mexico City 04510, Mexico; manlio.marquez@gmail.com; 3Department of Immunology, National Institute of Cardiology Ignacio Chávez, Mexico City 04510, Mexico; mireya.martinez@cardiologia.org.mx (M.M.-G.); malinalli.brianza@cardiologia.org.mx (M.B.-P.); 4Computational Genomics Division, National Institute of Genomic Medicine, Mexico City 14610, Mexico; 5Center for Complexity Sciences, Universidad Nacional Autónoma de México, Mexico City 04510, Mexico

**Keywords:** uric acid, regression-based machine learning, feature selection, feature engineering, Mexico City, Tlalpan 2020 cohort

## Abstract

**Background/Objectives:** Uric acid is a key metabolic byproduct of purine degradation and plays a dual role in human health. At physiological levels, it acts as an antioxidant, protecting against oxidative stress. However, excessive uric acid can lead to hyperuricemia, contributing to conditions like gout, kidney stones, and cardiovascular diseases. Emerging evidence also links elevated uric acid levels with metabolic disorders, including hypertension and insulin resistance. Understanding its regulation is crucial for preventing associated health complications. **Methods:** This study, part of the Tlalpan 2020 project, aimed to predict uric acid levels using advanced machine learning algorithms. The dataset included clinical, anthropometric, lifestyle, and nutritional characteristics from a cohort in Mexico City. We applied Boosted Decision Trees (Boosted DTR), eXtreme Gradient Boosting (XGBoost), Categorical Boosting (CatBoost), and Shapley Additive Explanations (SHAP) to identify the most relevant variables associated with hyperuricemia. Feature engineering techniques improved model performance, evaluated using Mean Squared Error (MSE), Root-Mean-Square Error (RMSE), and the coefficient of determination (R^2^). **Results:** Our study showed that XGBoost had the highest accuracy for anthropometric and clinical predictors, while CatBoost was the most effective at identifying nutritional risk factors. Distinct predictive profiles were observed between men and women. In men, uric acid levels were primarily influenced by renal function markers, lipid profiles, and hereditary predisposition to hyperuricemia, particularly paternal gout and diabetes. Diets rich in processed meats, high-fructose foods, and sugary drinks showed stronger associations with elevated uric acid levels. In women, metabolic and cardiovascular markers, family history of metabolic disorders, and lifestyle factors such as passive smoking and sleep quality were the main contributors. Additionally, while carbohydrate intake was more strongly associated with uric acid levels in women, fructose and sugary beverages had a greater impact in men. To enhance model robustness, a cross-feature selection approach was applied, integrating top features from multiple models, which further improved predictive accuracy, particularly in gender-specific analyses. **Conclusions:** These findings provide insights into the metabolic, nutritional characteristics, and lifestyle determinants of uric acid levels, supporting targeted public health strategies for hyperuricemia prevention.

## 1. Introduction

Uric acid is a widely studied biomarker due to its association with various health conditions, including gout, hypertension, and cardiovascular diseases. When uric acid levels exceed the reference range in the blood, the condition is termed hyperuricemia (HU). This condition can be a prevalent metabolic disorder. HU not only increases the risk of developing gout but also serves as a significant risk factor for cardiovascular diseases, which are the leading cause of death in Mexico and worldwide. Monitoring uric acid levels is crucial because it allows for the early detection of HU, helping to prevent the onset of gout and reduce the risk of kidney stones and cardiovascular complications [1,2].

The application of machine learning techniques has significantly influenced the detection and identification of elevated uric acid levels. By analyzing large datasets containing clinical and biochemical parameters, machine learning models were able to predict the likelihood of hyperuricemia with greater accuracy than traditional methods [3,4].

In a study led by Sangwoo Lee et al. [5], key performance metrics were used to evaluate the effectiveness of various machine learning algorithms in predicting HU. The Naïve Bayes algorithm achieved the highest sensitivity at 0.73, making it the most sensitive in detecting elevated uric acid levels. The Random Forest also performed well, with a sensitivity of 0.66 and the highest Balanced Classification Rate at 0.68. Additionally, RFC demonstrated superior performance in overall prediction accuracy with an Area Under the Curve of 0.775, compared to 0.669 for Naive Bayes and 0.568 for logistic regression models. Likewise, in a study by Zeng et al. [6], a stacked multimodal machine learning model was developed to predict HU by integrating genetic and clinical data. The model achieved high accuracy, with an AUC of 0.859 in the training set, effectively stratifying individuals into high and low risk groups for gout and other metabolic disorders. Additionally, the study found that lifestyle changes could mitigate adverse outcomes in high-risk individuals.

Conversely, another study by Sampa et al. [7] aimed to develop a predictive model for blood uric acid levels using data from health checkups, nutritional information, and socio-demographic characteristics to apply machine learning regression algorithms, including Boosted DTR, Decision Forest Regression, Neural Network, Bayesian Linear Regression, and Linear Regression. Among these, the Boosted DTR model demonstrated the best performance, achieving a notably low RMSE of 0.03.

These studies, along with others in the field [8,9,10], underscore the potential of machine learning approaches in accurately predicting uric acid levels. This evidence justifies the application of machine learning for early intervention and cost-effective management of related health conditions. While classification algorithms are typically used to categorize patients into groups or classes, such as normal or elevated uric acid levels, regression algorithms are particularly useful when predicting the exact value of uric acid concentration. In the context of the Tlalpan 2020 project, an analysis was conducted to develop a predictive model for identifying elevated uric acid levels using machine learning regression algorithms, including Boosted DTR, XGBoost, CatBoost, and SHAP. The approach involves applying these algorithms to identify the most relevant features influencing uric acid levels. While previous research has demonstrated performance variations across different machine learning models in metabolic studies, this study adopted an exploratory approach to compare multiple algorithms and identify the most effective predictors of uric acid levels. Given the variability in machine learning model performance across different datasets and study populations, this study did not assume a predefined hypothesis regarding the superiority of any particular model. Instead, an empirical evaluation was conducted to determine which approach performed best under the conditions of the dataset used. After each algorithm generates its respective subset of important features, a cross-feature analysis was performed to combine the top features from each algorithm. This process aimed to enhance the model’s performance by leveraging the strengths of multiple feature sets, leading to more accurate predictions.

## 2. Materials and Methods

### 2.1. Data

This study used baseline data from the Tlalpan 2020 cohort at the National Institute of Cardiology Ignacio Chávez in Mexico City [11]. Although the Tlalpan 2020 cohort was designed as a longitudinal study, this analysis focused on baseline data, as it provided the most complete and consistent set of variables across participants. This cross-sectional approach allowed us to explore initial associations between uric acid levels, nutritional intake, and lifestyle factors before assessing long-term changes. The study was approved by the institution’s Bioethics Committee under code 13-802, and the research involved informed and consenting volunteers. The Tlalpan 2020 project was established to identify risk factors for hypertension and other cardiometabolic conditions. Participants, aged 20 to 50 years and clinically healthy at recruitment, have been followed biennially since 2014. The dataset includes clinical, anthropometric data, alcohol and tobacco use, physical activity, economic income, education, family health history, biomedical evaluations, sleep quality, and nutrient intake.

Sleep quality was measured using the MOS Sleep Scale [12], which evaluates sleep through 12 questions on interruptions, snoring, breathing difficulties, headaches on waking, overall sleep quality, and daytime drowsiness, and tracks total sleep hours over the previous four weeks. Clinical and anthropometric data, such as systolic and diastolic blood pressure (SBP and DBP), were collected following the procedure outlined in [13], with waist circumference, height, and weight measured according to ISAK guidelines [14]. Biochemical assessments were conducted using blood and urine samples to analyze various metabolic and lipid markers. Data on lifestyle factors, including smoking, alcohol intake, and physical activity, were gathered, with physical activity levels assessed using the International Physical Activity Questionnaire [15] and categorized as low, moderate, or high based on metabolic equivalent minutes per week. Educational attainment was recorded as primary, secondary, or tertiary (including postgraduate) levels, and economic income levels were segmented into low, medium, and high categories based on monthly income in Mexican pesos. Nutritional habits, including daily meal frequency and nutrient intake over the past year, were tracked and analyzed using the “Evaluation of Nutritional Habits and Nutrient Consumption System” software tool Version 2 [16].

All data mentioned in this section are presented in Appendix A, which is available in Appendix A, where the letter D stands for dichotomous and the letter C stands for continuous.

### 2.2. Methods

#### 2.2.1. Machine Learning Approaches

We applied four machine learning algorithms in this study: XGBoost, Boosted DTR, CatBoost, and SHAP. XGBoost was developed by Tianqi Chen [17], while Boosted DTR was pioneered by Robert E. Schapire and Yoav Freund [18,19]. CatBoost was created by Yandex [20]. Finally, SHAP, introduced by Lundberg and Lee [21], was used for feature selection due to its proven effectiveness in diverse fields such as healthcare.

XGBoost is known for its speed and performance on large datasets, Boosted DTR combines multiple decision trees for a robust predictive model, CatBoost stands out for its remarkable efficiency and performance, particularly excelling in regression problems. SHAP is employed for feature selection due to its demonstrated effectiveness across diverse domains, such as healthcare [22]. It offers valuable insights into the contribution of each feature to the model’s predictions.

#### 2.2.2. Feature and Cross-Feature Selection

Feature selection is crucial in machine learning for enhancing model efficiency and accuracy, particularly in regression algorithms. It helps identify key predictors, reduces overfitting, and improves computational efficiency. After identifying these key features, cross-feature selection can be employed to further refine the model by evaluating the interactions between features across multiple datasets or folds. This process evaluates the interactions between features from multiple datasets or models, which is essential for uncovering complex relationships between variables.

Although cross-feature selection offers a promising approach to uncovering complex interactions between variables in predictive models, its adoption remains relatively limited compared to more conventional methods. However, the cross-feature selection method has shown significant results in studies related to cancer and various diseases [23]. These investigations have demonstrated that this approach can enhance the accuracy of predictive models and facilitate the identification of complex interactions that other conventional methods might overlook.

In this study, cross-feature selection was applied as follows:

Feature subsets (in this case 3), F_1, F_2, and F_3, were derived from the three best performing models. The top 5 features from each subset were then selected for further analysis. Finally, the common features among F_1, F_2, and F_3 were identified to enhance the model’s robustness, as represented by the following equation:(1)$$F_{\mathrm{cross}}=\left\{X_{1,i}\times X_{2,j}\times X_{3,k}|i,j,k\in \left\{1,2,3,4,5\right\}\right\}$$

This approach ensured that the final model incorporated the most influential features from each of the top-performing models, improving its overall performance.

Overfitting was controlled by performing cross-validation and hyperparameter tuning to ensure model generalization. Additionally, feature selection was based on multiple algorithms (Boosted DTR, CatBoost, SHAP, and XGBoost), reducing irrelevant or redundant variables while preserving the most informative ones. Although some selected features may be intercorrelated, machine learning models such as XGBoost and CatBoost can capture non-linear relationships and interactions among predictors, which conventional statistical models may overlook.

#### 2.2.3. Performance Measures

We used the metrics MSE, RMSE, and $R^{2}$, which are employed in regression machine learning algorithms, to assess model performance.

The MSE is calculated as the sum of the squares of the differences between the model predictions ($\hat{y}$) and the actual values (*y*), divided by the number of samples (*n*), expressed by the equation:(2)$$MSE=\frac{1}{n}\sum_{i=1}^{n}{\left(y_{i}-{\hat{y}}_{i}\right)}^{2}$$

The RMSE, the square root of the MSE, is obtained by taking the square root of the MSE value, providing an error metric on the same scale as the original data, facilitating the interpretation of the prediction error.

On the other hand, $R^{2}$ is calculated as the proportion of variance explained by the model relative to the total variance of the data, represented by the equation:(3)$$R^{2}=1-\frac{\sum_{i=1}^{n}{\left(y_{i}-{\hat{y}}_{i}\right)}^{2}}{\sum_{i=1}^{n}{\left(y_{i}-\bar{y}\right)}^{2}}$$

### 2.3. Experimental Setup and Process

All experiments were performed using Python 3.11 and Jupyter Notebook version 7. Min–max scaling from the scikit-learn library [24] was used to normalize continuous variables, and dichotomous variables were represented as numbers.

The overall experimental process is illustrated in Figure 1. It began with the extraction of relevant variables, followed by data cleaning and normalization using the min–max method. Once the dataset was prepared, various models were executed, specifically XGBoost, Boosted DTR, SHAP, and CatBoost. During the training process, parameter tuning was performed to optimize the performance of each model. Subsequently, feature selection was applied to identify the most relevant variables for each algorithm, grouping them into specific subsets.

With the subsets defined, the performance of each algorithm was analyzed using the MSE, RMSE, and $R^{2}$ metrics. To implement the cross-feature selection method, the five most important features from each subset were selected, considering the corresponding group and sex. These five variables were then combined using the XGBoost algorithm, which was chosen for its high performance in regression problems, computational efficiency, compatibility with SHAP, and control over overfitting.

After obtaining the cross-feature selection results and confirming that the metrics were higher than those of the other algorithms used, the most important variables from each subgroup were presented in a spiderweb graph.

## 3. Results

In this section, we present the results derived from applying Boosted DTR, CatBoost, SHAP, and XGBoost to the dataset, which was initially divided into the following groups: clinical, anthropometric, and lifestyle variables; food frequency questionnaire data; and nutritional and biochemical features obtained from the food frequency questionnaire. The results were segmented by sex, dividing the analysis between men and women. This separation is crucial to identify the factors that influence each group differently, considering sex as a variable that can modify the relationship between risk factors and health outcomes. This approach allowed for a more precise and nuanced understanding of how different factors specifically impact the health of men and women.

### 3.1. Clinical, Anthropometric, and Lifestyle Variables

Table 1 illustrates how the variables were identified and evaluated based on their relative importance in predicting health outcomes in women. Algorithms utilizing the variable importance metric, such as Boosted DTR and CatBoost, highlighted Body Mass Index (BMI), triglycerides, and creatinine as the most influential factors. On the other hand, SHAP and XGBoost used the mean absolute SHAP value to measure feature contribution, with BMI, triglycerides, and creatinine also standing out as key contributors. Moreover, the subset of features identified by XGBoost shows some unique differences compared to the other models. Variables such as passive smoking, having had a baby weighing more than 4 kg (BB4K), mother with diabetes, and snoring were highlighted by XGBoost as important factors. These features, particularly those related to lifestyle and family health history, suggest that XGBoost is capturing nuances in health outcomes that might be less prominent in the other models. This indicates that, in addition to clinical and anthropometric factors, elements of personal and familial health history play a significant role in predicting health outcomes in women, as identified by XGBoost.

For women, Table 2 shows the performance metrics obtained by each algorithm (CatBoost, Boosted DTR, SHAP, and XGBoost), when applied to the subset of Clinical, Anthropometric, and Lifestyle Features. Each algorithm identifies different sets of important features, and these features were used to evaluate model performance through metrics such as MSE, RMSE, and $R^{2}$.

Based on the performance metrics, the top three algorithms for predicting uric acid levels in women were XGBoost, SHAP, and CatBoost. XGBoost stood out as the best performer with the lowest MSE (0.0079) and RMSE (0.0890), and the highest $R^{2}$ value (0.3170), indicating both superior predictive accuracy and the ability to explain the most variance in the data. SHAP followed closely, with strong predictive accuracy reflected in its low MSE (0.0068) and RMSE (0.0824), although its $R^{2}$ (0.1688) was slightly lower, meaning it explained less variance compared to XGBoost. CatBoost, while having the lowest MSE (0.0063) and RMSE (0.0793), ranked third overall due to its $R^{2}$ value of 0.2315, placing it between XGBoost and SHAP in terms of variance explanation. Boosted DTR performed reasonably well, with an MSE of 0.0087 and an RMSE of 0.0930, indicating it was slightly less accurate than XGBoost, SHAP, and CatBoost.

In the case of men, Table 3 illustrates a similar approach highlighting BMI, waist circumference, creatinine, atherogenic index, triglycerides, and being a daily smoker as the most critical factors. However, SHAP and XGBoost revealed additional nuances in the subset of features important for men’s health. For instance, XGBoost identified passive smoking, father with gout, mother with diabetes, and alcoholic beverages such as whiskey and cognac consumption as significant contributors. Moreover, the presence of high cholesterol and the type of health insurance (IMSS) were also emphasized. These findings suggest that, similar to women, a mix of clinical, anthropometric, and lifestyle factors plays a vital role in determining health outcomes in men. Yet, XGBoost’s ability to capture lifestyle and family health history nuances offers a deeper understanding of how these elements uniquely impact men’s health outcomes.

In terms of performance metrics (see Table 4), the top three algorithms based on performance metrics were XGBoost, Boosted DTR, and SHAP. XGBoost led with the highest $R^{2}$ value of 0.1996, although it had a slightly higher MSE (0.0137) and RMSE (0.1072) compared to the other models. Boosted DTR followed closely with a lower MSE (0.0111) and RMSE (0.1054), and an $R^{2}$ of 0.1994. SHAP was very similar to Boosted DTR, with identical MSE and RMSE values but a slightly lower $R^{2}$ of 0.1987.

### 3.2. Food Frequency Questionnaire Features

Table 5 presents the subset of features derived from the food frequency questionnaire for men. In this case, the Boosted DTR algorithm highlighted foods and beverages that could have a significant impact on uric acid levels.

In the case of CatBoost, medium-sized cola was again highlighted, along with corn tortilla and a portion of pork. Additionally, chocolate powder, white sliced bread, atole with milk, and a bowl of oatmeal were emphasized as key elements in the diet that could contribute to increased uric acid levels.

The SHAP analysis also underscored the importance of a medium-sized cola and a cup of atole with milk. It is interesting to note that banana consistently appeared in all three algorithms. Although generally considered healthy, this fruit is rich in fructose, which could justify its appearance as a predictor in these models, given that fructose can contribute to increased uric acid levels. Additionally, SHAP highlights foods similar to those identified by other algorithms such as margarine, a portion of pork, chorizo, and sweet bread, which are integral parts of the diet and may influence uric acid levels.

Table 6 presents the performance metrics, where Boosted DTR and CatBoost showed very similar performance, both with an MSE of 0.0111 and RMSE values of 0.1051 and 0.1052, respectively. Boosted DTR had a slightly higher $R^{2}$ value (0.0767) compared to CatBoost (0.0763), indicating that it explained marginally more variance, making it the better of the two. SHAP outperformed both in terms of predictive accuracy, as evidenced by its lower MSE (0.0083) and RMSE (0.0913). However, SHAP’s $R^{2}$ value was significantly lower at 0.0187, suggesting it explained much less variance in the data, potentially indicating that while SHAP is good at making accurate predictions, it may not capture the underlying patterns as effectively as the other models.

Concerning women, several nutritional risk factors were associated with elevated uric acid levels. A medium-sized cola was particularly concerning due to its high sugar content. Hard liquor, such as rum, brandy, or tequila, was identified as a risk factor for increasing uric acid levels. Carnitas, a dish high in purines due to its pork content, was another noteworthy risk factor. Rice dishes, particularly those with a high glycemic index, can elevate blood glucose levels, potentially impacting uric acid metabolism negatively.

Sugary drinks, like sodas and other sweetened beverages, were also flagged as a concern. Butter is traditionally considered beneficial due to its healthy fats; however, when consumed in combination with high-carbohydrate foods or if sourced from less healthy origins, it can contribute to inflammation and exacerbate uric acid levels. The consumption of safflower oil, which is high in omega-6 fatty acids, could be problematic if consumed in excess, as it may promote inflammation, a factor associated with higher uric acid levels. Lastly, wheat tortillas were identified as a risk factor for increased uric acid levels.

Table 7 shows the performance metrics for CatBoost, Boosted DTR, and SHAP on the food frequency questionnaire features for women. CatBoost and Boosted DTR performed similarly, both with an MSE of 0.0111 and RMSE around 0.105, and their $R^{2}$ values were close, with Boosted DTR slightly ahead. SHAP had a lower MSE (0.0083) and RMSE (0.0913), indicating better accuracy, but its $R^{2}$ was much lower (0.0187), meaning it explained less variance in the data. Overall, CatBoost and Boosted DTR offered a better balance between accuracy and variance explanation.

### 3.3. Nutritional and Biochemical Features

Table 8 presents a subset of nutritional and biochemical features identified in both women and men, using only the CatBoost and SHAP algorithms. These algorithms were chosen due to their outstanding performance with this dataset.

For women, Table 8 shows that CatBoost highlighted glucose as the most important feature, followed by carbohydrates and alcohol. Other important features included maltose, fructose, and cholesterol, which further emphasized the metabolic and nutritional components potentially affecting uric acid levels. SHAP also prioritized glucose, consistent with CatBoost, and additionally underscored the significance of beta-tocopherol and beta-carotene. Alcohol and cholesterol were also highlighted by SHAP, reinforcing the importance of nutritional and lifestyle factors.

Regarding the metrics obtained by the nutrient algorithms in women, SHAP stood out with the lowest error metrics (MSE of 0.0077 and RMSE of 0.0880), indicating it was the most accurate model. However, CatBoost had a higher $R^{2}$ value of 0.1139, suggesting it explained more variance in the data. While SHAP was more accurate in its predictions, CatBoost provided a broader understanding of the data variance, making it a strong contender for modeling nutritional and biochemical features in women.

In the case of men, Table 8 shows that CatBoost identified fructose as the most influential feature, followed closely by maltose and glucose. Additional significant features included carbohydrates, lactose, and alcohol, indicating that a combination of sugars and alcohol consumption may be pivotal in understanding men’s uric acid metabolism.

SHAP similarly highlighted fructose and glucose, both sugars strongly linked to increased uric acid levels. Additionally, it emphasized cholesterol, whose association with lipids and cardiovascular health may indicate a connection with HU. Lactose and carbohydrates were also key features identified by SHAP. Additionally, maltose was highlighted, further emphasizing the role of specific sugars in influencing uric acid levels.

Based on Table 9 for men, the SHAP model demonstrated a better ability to explain the variance in the data with an $R^{2}$ of 0.1345, despite having slightly higher error metrics (MSE of 0.0135 and RMSE of 0.1160) compared to CatBoost. CatBoost, with an $R^{2}$ of 0.0654, showed a lower capacity to explain the variance, but it had slightly better error metrics (MSE of 0.0130 and RMSE of 0.1139). Overall, SHAP provided a better understanding of the underlying data patterns, while CatBoost offered marginally better accuracy in predictions.

### 3.4. Results of Cross-Feature Selection

Once the feature subsets were obtained from the XGBoost, CatBoost, Boosted DTR, and SHAP models, it was observed that some models had similar metric results. In the case of the anthropometric variables group for both men and women, XGBoost identified different variables compared to Boosted DTR, CatBoost, and SHAP, which captured more variables related to lifestyle, habits, and family medical history. For women, in Table 2, XGBoost stood out as the best algorithm, showing superior metrics in terms of precision and data variability. However, the subset of features selected by XGBoost (see Table 1) excluded important variables such as triglycerides, creatinine, weight, the atherogenic index, and other potentially significant variables. For this reason, the cross-feature selection method was applied, considering the top five variables from each subset obtained by each algorithm in each group, segmented by sex.

For women, the clinical, anthropometric, and lifestyle variables group was analyzed using the CatBoost, SHAP, and XGBoost algorithms, as they demonstrated the highest performance. After removing duplicate features, the top 10 unique variables identified were triglycerides, BMI, creatinine, atherogenic index, urinary sodium, mother with diabetes, passive smoker, baby birth weight over 4 kg, snoring, and glucose. Similarly, for men, the same algorithms were selected based on their performance, and the top features identified were creatinine, BMI, atherogenic index, triglycerides, urinary sodium, father with diabetes, passive smoker, father with gout, mother with diabetes, and weight.

In the food frequency group, data for women were analyzed using the Boosted DTR, CatBoost, and SHAP algorithms, as they demonstrated the highest performance. The top five variables identified were a medium cola soda, a teaspoon of butter, a drink (rum, brandy, or tequila), a teaspoon of hot sauce or chili, a portion of carnitas, a glass of sugary flavored water, a cup of coffee without sugar, a teaspoon of margarine, a bowl with rice, and safflower oil. Likewise, the essential features identified for men were a medium cola soda, a corn, safflower oil, a teaspoon of margarine, corn oil, a banana, a portion of pork, a teaspoon of chocolate powder, a cup of atole with milk, and chicken egg.

The nutritional and biochemical features for women were analyzed using the CatBoost and SHAP algorithms, which showed the best results (see Table 8). The top five variables identified included glucose, carbohydrates, alcohol, fructose, maltose, total cholesterol, starch, and beta cryptoxanthin. Similarly, the essential features identified for men were fructose, maltose, glucose, carbohydrates, lactose, total cholesterol, alcohol, and beta cryptoxanthin.

Table 10 presents the performance metrics (MSE, RMSE, and $R^{2}$) for the cross-feature method, applied using XGBoost to the clinical, anthropometric, lifestyle, food frequency questionnaire, and nutritional and biochemical subsets across both sexes. By combining the top five features from each subgroup, this method consistently outperformed individual algorithms, demonstrating superior predictive performance across all metrics and groups.

To effectively present the features used in the cross-feature selection for each group and sex, radar charts were created (see Figure 2, Figure 3 and Figure 4). Given that the scale of the importance method used by Boosted DTR and CatBoost differed from the MASV method applied by SHAP and XGBoost, it was necessary to implement a series of normalization steps. The min–max normalization technique was applied to scale the importance values within a range of [0, 1], adjusting the variables to a common scale. This ensured that the importance assigned by each algorithm was reflected in a coherent and comparable manner. After normalization, the data for each gender were merged, and the mean of the normalized values was calculated to obtain a representative importance score for each variable.

To further evaluate the stability of the cross-feature selection method, we performed a sensitivity analysis varying max_iter, max_depth, and l2_regularization. Our original analysis was conducted using XGBoost, CatBoost, and Boosted DTR; however, to further validate the robustness of our approach, we applied HistGradientBoostingRegressor for this sensitivity analysis. This method was selected due to its efficiency in handling structured tabular data and its ability to assess model stability under different hyperparameter settings. The results (Figure 5 and Figure 6) confirm that increasing max_iter did not significantly improve predictive performance. Additionally, deeper models tended to perform worse, likely due to overfitting and increased variance, which reduced generalization. These findings suggest that the cross-feature selection method maintained stable performance across different hyperparameter configurations, indicating robustness. The complete numerical results of the sensitivity analysis are provided in Appendix A.

## 4. Discussion

This study offers a comprehensive examination of the factors influencing uric acid levels in the Tlalpan 2020 cohort, with a particular focus on gender-specific differences. In this regard, it is relevant to notice that women constituted a smaller proportion of the cohort, which likely reduced the variability in their data, leading to more accurate predictions. Additionally, the predictors for women might not have carried as much weight, as some variables may not have been as representative or did not capture the full variability in the cohort. This could have resulted in more stable model performance but with a lesser predictive impact. To further improve the algorithms, it would be crucial to continue collecting more data, which could enhance the representation of key variables and ultimately improve the models’ predictive power. Utilizing machine learning models, including Boosted DTR, XGBoost, CatBoost, and SHAP, we identified critical variables across clinical, anthropometric, lifestyle, and nutritional domains. Additionally, the cross-feature selection method was employed to refine the model further by combining the top features identified by each algorithm. To effectively analyze and compare these variables, spiderweb charts were used, providing a clear visualization of the interactions between variables and enhancing the understanding of the predictive factors across different groups.

The degree of bias in cohort populations when generalizing the results observed in the Tlalpan 2020 study depends on several factors, particularly the representativeness of the participants relative to the general population. One key limitation is the age restriction, as all participants were younger than 50 years at baseline, which may prevent the findings from being extrapolated to older populations where the relationship between uric acid and health outcomes differs, such as the observed uric acid paradox in the elderly. Additionally, geographical and socioeconomic factors play a role, as the cohort consisted of residents of Mexico City, potentially limiting the applicability of the results to populations with different nutritional habits, genetic backgrounds, and environmental exposures. Selection bias is another concern, as individuals who chose to participate may have had distinct health statuses, lifestyles, or access to healthcare, making them non-representative of the broader population at risk of hyperuricemia-related conditions. To mitigate these biases, sensitivity analyses, external validation in diverse populations, and age-stratified modeling approaches may enhance the robustness and applicability of the findings.

### 4.1. Key Clinical, Anthropometric and Lifestyle Predictors

Figure 2 highlighted the different health, hereditary, and lifestyle factors contributing to elevated uric acid levels in men and women.

In men, BMI was a critical factor, indicating that higher body weight and obesity are closely linked to elevated uric acid levels, likely due to insulin resistance impairing uric acid excretion. Age was another important factor, with older men showing a higher risk of elevated uric acid levels due to declining renal function and increased comorbidities. Elevated creatinine, a marker of kidney function, suggested that impaired renal function was a key contributor to HU in men. Alcohol consumption, as indicated by the cognac factor, was notably associated with uric acid levels. Additionally, a hereditary factor such as having a father with gout was significant, highlighting a genetic predisposition to HU. These factors together suggest that managing weight, addressing metabolic health, moderating alcohol intake, and considering hereditary risk are vital strategies for controlling uric acid levels in men.

In women, BMI was a significant factor, emphasizing the role of obesity in the development of HU. Snoring also emerged as important, with poor sleep linked to increased stress and inflammation, both of which can elevate uric acid levels. Additionally, the atherogenic index was significant, as it reflected the relationship between lipid profiles and cardiovascular risk, with a higher index being associated with increased uric acid levels and a greater risk of cardiovascular disease. High cholesterol further complicates the picture, as dyslipidemia is associated with higher uric acid levels and cardiovascular risks. Hereditary factors also played a role, with a mother’s diabetes and a father’s diabetes both contributing to a genetic predisposition to metabolic conditions that may elevate uric acid levels. Finally, exposure to passive smoking was a significant lifestyle factor, as it could increase oxidative stress and inflammation, potentially exacerbating uric acid levels.

### 4.2. Food Frequency Questionnaires Factors

Figure 3 revealed significant nutritional patterns among men and women, highlighting specific food items strongly associated with the risk of elevated uric acid levels.

Specifically, the consumption of medium-sized cola in men and women was a significant nutritional factor influencing uric acid levels, aligning with existing evidence linking sugary beverages, particularly those high in fructose, to elevated uric acid [25,26]. Fructose metabolism in the liver is crucial in uric acid production, and frequent consumption of such beverages could exacerbate HU.

Figure 3 identified several other nutritional factors that were particularly relevant in men, including the consumption of corn tortillas, corn, pork, and tamales. Corn and corn-based products, such as tortillas, are staples in many diets but are high glycemic index foods, which can lead to rapid spikes in blood glucose and exacerbate insulin resistance, thereby reducing the body’s ability to effectively excrete uric acid. Pork, being rich in purines, directly contributes to increased uric acid levels, raising the risk of gout and related conditions. Tamales, which combine high glycemic corn masa and purine rich pork, represent a potent nutritional source of elevated uric acid levels [27], further compounded by the use of fats like lard in their preparation. These factors suggest that men who frequently consume these traditional foods might benefit from nutritional modifications, such as reducing the intake of high-purine and high-glycemic index foods and incorporating more low-glycemic vegetables, fruits, and lean proteins to manage uric acid levels effectively.

In women, several other nutritional factors identified were a teaspoon of butter, a portion of carnitas, and rice dishes, which are associated with elevated uric acid levels. Butter, while rich in beneficial fats and potentially healthful when consumed in moderation, can contribute to increased uric acid levels, especially when frequently combined with high-carbohydrate foods like rice [28]. Additionally, carnitas, a dish high in purines due to its pork content, directly contributes to uric acid production [27]. Therefore, while foods like butter may offer health benefits, their consumption alongside carbohydrate-heavy foods should be moderated to prevent exacerbating uric acid levels.

### 4.3. Nutritional and Biochemical Factors

This study provided an analysis of the nutritional and biochemical factors influencing uric acid levels, with a particular focus on gender differences. The findings revealed that while some factors, such as glucose and cholesterol, were common between men and women, others exhibited gender-specific influences, suggesting the need for personalized approaches in the prevention and management of HU.

Fructose was identified as a key factor in men, aligning with the existing literature that links high fructose consumption with increased uric acid production [29,30,31]. Through its metabolism in the liver, fructose leads to the generation of urate, thereby exacerbating uric acid levels in the blood. It is important to note, however, that the impact of fructose on uric acid levels can vary significantly depending on the source and type of fructose consumed. For instance, fructose found in whole fruits is accompanied by fiber, vitamins, and antioxidants, which may mitigate some of its negative effects. In contrast, the fructose in processed foods, such as commercial baked goods, is often present in a more concentrated form and combined with other refined sugars, leading to a more pronounced increase in glucose and urate levels.

In addition to fructose, maltose and lactose also emerged as notable risk factors, specifically in men. Maltose, commonly found in malted products and certain grains, contributes to uric acid elevation through mechanisms similar to those of fructose, by promoting rapid glucose spikes and potentially exacerbating insulin resistance, which can impair uric acid excretion. Similarly, lactose, primarily found in dairy products, was identified as a significant factor, suggesting that men who consume high amounts of dairy may also be at increased risk of elevated uric acid levels.

Understanding the type of fructose consumed by the population, whether simple or complex, is crucial for tailoring nutritional recommendations. Moreover, it is essential to educate the population about the maximum recommended intake of fructose, particularly from added sugars, which should ideally be limited to no more than 10% of daily caloric intake [32] according the World Health Organization. This distinction underscores the importance of not only reducing overall fructose intake but also making informed choices about the sources of fructose, which can significantly influence the risk of developing HU.

Cholesterol was another significant factor in men, consistent with studies in the literature [33,34] that suggest a complex interaction between lipid metabolism and uric acid. Elevated cholesterol levels can contribute to systemic inflammation and oxidative stress, both of which are linked to increased uric acid levels. This finding highlights the need to integrate cholesterol management as part of a broader approach to controlling HU in the male population.

In contrast, carbohydrates and glucose emerged as an influential factor in women. This observation is supported by studies in the literature that have demonstrated a strong connection between high carbohydrate intake, particularly from foods with a high glycemic index, and elevated blood glucose levels [35,36]. Increased glucose levels can lead to insulin resistance, a condition that diminishes the kidneys’ capacity to excrete uric acid efficiently. Research indicates that diets high in high glycemic carbohydrates not only cause significant blood glucose spikes but also aggravate conditions such as metabolic syndrome, which is closely associated with HU [37]. These findings highlight the critical role of glycemic control in nutritional interventions aimed at managing uric acid levels, particularly in women who may be more vulnerable to these metabolic effects [38].

Additionally, it is observed that women consume foods containing beta-tocopherol and beta-carotene, which are antioxidants known for their protective role against oxidative stress [39,40], a factor that contributes to the accumulation of uric acid. Vitamin E, particularly in its various forms including beta-tocopherol, and beta-carotene, as a precursor to vitamin A, also function as antioxidants. However, as seen in Figure 4, the intake of these antioxidants was relatively low. This suggests that although women are including these nutrients in their diet, their consumption may not be sufficient to fully counteract the negative effects of a high carbohydrate intake on uric acid levels.

While machine learning models such as XGBoost and CatBoost improve predictive accuracy, their complexity can limit interpretability. To address this, we leveraged SHAP values to quantify feature importance and enhance model transparency. In this study, we applied SHAP to identify relationships among the most relevant variables and explore sex-specific differences. Additionally, we generated dependency plots and clustering-based SHAP analyses to visualize distinct risk patterns, strengthening the interpretation of our results.

## 5. Conclusions

This study, framed within the Tlalpan 2020 project, demonstrated the effectiveness of using machine learning models to predict uric acid levels in a diverse cohort from Mexico City. Through the implementation of algorithms such as Boosted DTR, XGBoost, CatBoost, and SHAP, key variables were identified across different categories, including clinical, anthropometric, lifestyle, and nutritional characteristics.

The findings highlighted that XGBoost was particularly effective in identifying critical factors in the anthropometric and clinical variables, while CatBoost excelled in analyzing the food frequency questionnaire, and SHAP provided detailed insights into the complex interactions between nutrients and uric acid levels. These tools not only improved predictive accuracy but also enhanced our understanding of the factors contributing to HU.

Furthermore, the application of the cross-feature selection method, in combination with spiderweb charts for visualization, proved crucial in refining the model and ensuring that the most significant variables were consistently identified and compared across different groups and genders. This approach facilitated a more nuanced analysis of the interactions between variables, highlighting the importance of gender-specific risk factors and nutritional patterns in managing uric acid levels.

The use of feature engineering techniques resulted in significant improvements in performance metrics, validating the importance of careful variable selection in predictive modeling. Moreover, the findings underscore the potential of machine learning models not just for prediction but also for uncovering complex relationships within large datasets, offering insights that could inform targeted interventions.

While the implementation of public health policies is beyond the scope of this study, our findings provide insights that could support future initiatives focused on early detection and prevention of hyperuricemia. Additionally, efforts are being considered to develop predictive models that enhance risk assessment and aid in data-driven decision-making.

This integrated approach, which considers gender differences and demographic factors, provides a strong foundation for future research and the development of personalized interventions. Ultimately, this work contributes significantly to the understanding of hyperuricemia and its public health implications. It also sets a precedent for the application of machine learning techniques in data analysis. The study also suggests practical pathways for the integration of data analytics into public health strategies, particularly in the prevention and management of conditions related to hyperuricemia.

The public health implications of the study could be further strengthened by specifying how the predictive modeling of uric acid levels can inform targeted interventions and policy recommendations. For instance, if machine learning models accurately identify nutritional and metabolic predictors of hyperuricemia, these insights could be used to develop personalized nutritional guidelines or community-based prevention programs aimed at reducing the burden of related conditions such as gout, kidney disease, and cardiometabolic disorders. Additionally, integrating these predictive tools into clinical practice could help healthcare professionals identify at-risk individuals earlier and implement lifestyle modifications before the onset of complications. Policymakers could also leverage these findings to design educational campaigns or refine nutritional recommendations tailored to specific demographic groups. By explicitly outlining these applications, the authors could provide a clearer connection between their findings and actionable public health strategies.

The results of this study could have valuable applications in clinical settings, particularly in cardiology clinics, where hyperuricemia is increasingly recognized as a potential risk factor for cardiovascular disease. If machine learning models can accurately predict uric acid levels based on metabolic and nutritional factors, clinicians could integrate these tools into routine assessments to identify patients at higher risk of developing conditions such as hypertension, atherosclerosis, or heart failure. This predictive capability would allow for earlier interventions, including lifestyle modifications, nutritional counseling, and, when necessary, pharmacological strategies to manage uric acid levels proactively. Incorporating these findings into risk models could improve personalized treatment by tailoring recommendations to individual metabolic profiles. By bridging predictive analytics with clinical decision-making, this research has the potential to improve preventive care and optimize cardiovascular risk management.

### Limitations

As every scientific research piece, the present work has some limitations; we can mention the following:Cohort-specific limitations: The study was based on data from the Tlalpan 2020 cohort, which represents a specific geographic and demographic group (Mexico City). The findings may not be generalizable to populations with different ethnic, socioeconomic, or geographic characteristics. A broader dataset might provide more generalizable results.Since the baseline participants of the Tlalpan 2020 cohort were younger than 50 years of age, the potential protective effect of uric acid observed in elderly populations (the *uric acid paradox*) might not be evident. Additionally, the non-linear relationship between uric acid and mortality within the 200–300 μmol/L range suggests that small variations in uric acid levels could have different health implications depending on the age and metabolic context of the population. This age-related effect should be considered when interpreting the predictive modeling outcomes, as the relationships observed in younger adults may not extrapolate directly to older populations.Potential bias in self-reported data: some data, such as lifestyle habits and nutritional intake from the food frequency questionnaire, are self-reported, which can introduce recall bias or under-reporting.Limited long-term data: This study focused on a snapshot of uric acid levels rather than on long-term changes. Uric acid concentrations can fluctuate daily and seasonally due to physiological and environmental factors; however, these variations were not captured in our cross-sectional analysis using baseline data. Additionally, individuals with a genetic predisposition to hyperuricemia might experience different fluctuation patterns, but this aspect was not assessed in our dataset.Limited exploration of confounding factors: while gender and lifestyle factors were considered, other confounders, such as genetic predispositions or concurrent medication use, might also impact uric acid levels and should be more thoroughly explored.Algorithm-specific limitations: Each machine learning algorithm has its limitations in terms of predictive accuracy and interpretability. For instance, models like CatBoost and XGBoost are powerful but complex, making it harder to interpret the relationships between features and outcomes. These limitations may hinder the clinical applicability of the findings.

## Figures and Tables

**Figure 1 nutrients-17-01052-f001:**
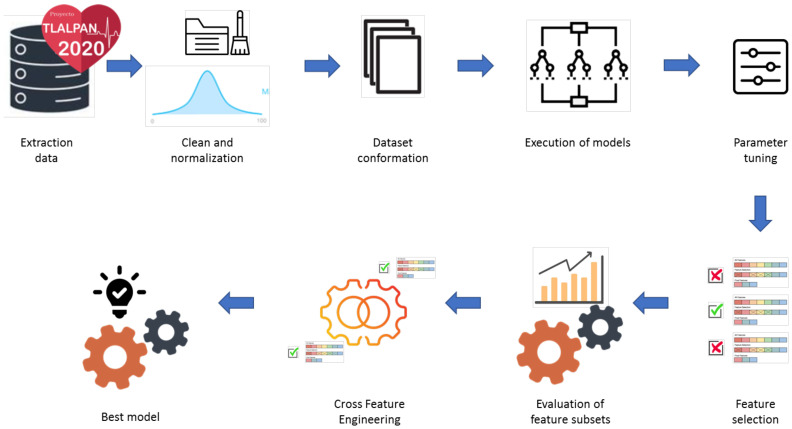
Experimental process.

**Figure 2 nutrients-17-01052-f002:**
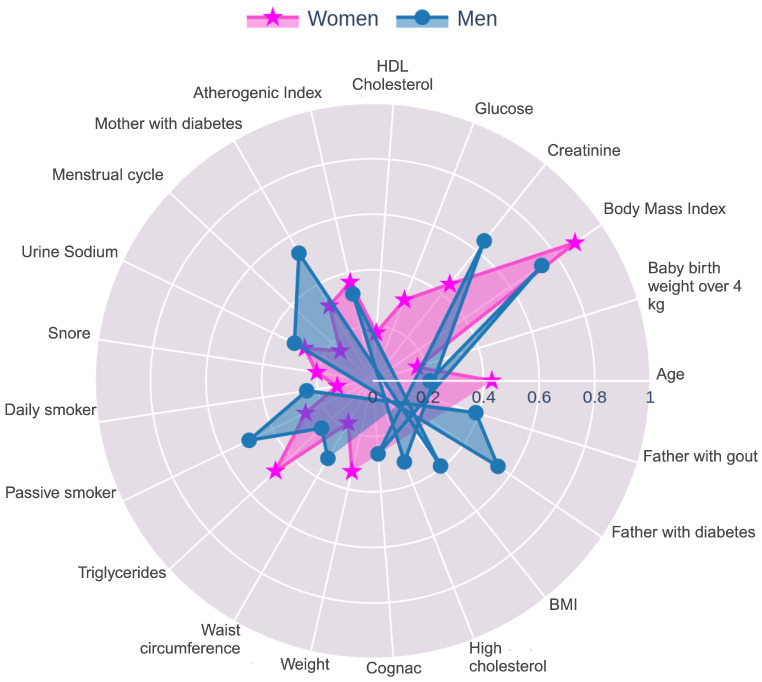
Radar chart of the most important clinical, anthropometric, and lifestyle features for women and men as identified in Table 1 and Table 3.

**Figure 3 nutrients-17-01052-f003:**
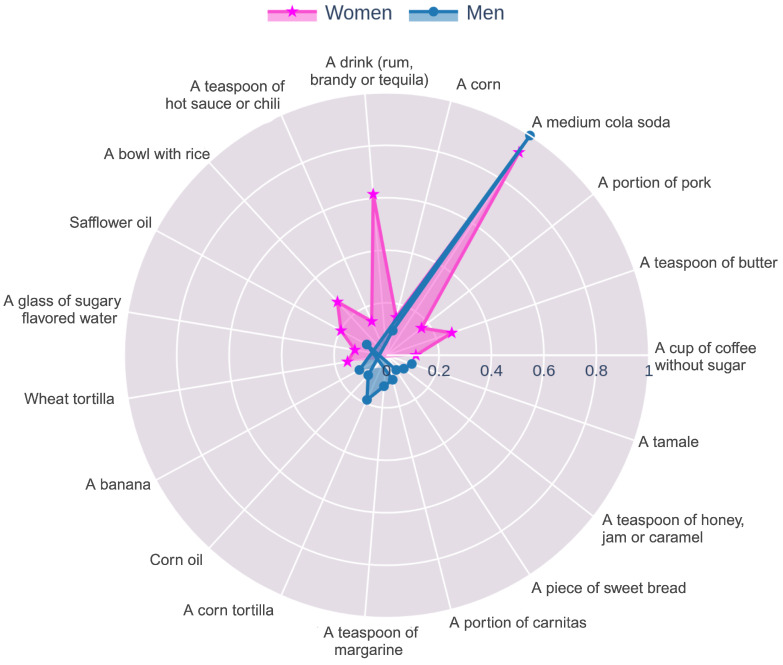
Radar chart of the most important features from a food frequency questionnaires for women and men as identified in Table 5 and Table 11.

**Figure 4 nutrients-17-01052-f004:**
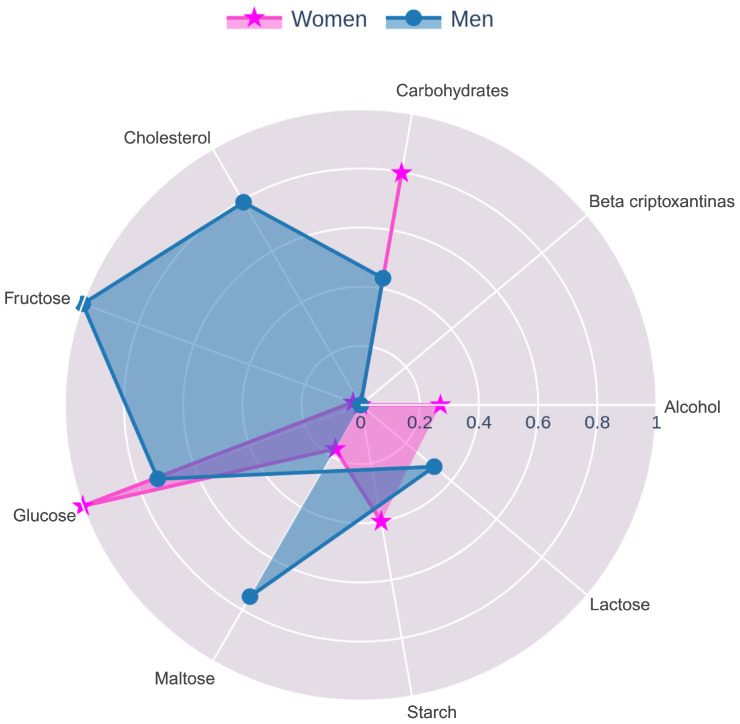
Radar chart of the most important nutritional and biochemical features for women and men as identified in Table 8.

**Figure 5 nutrients-17-01052-f005:**
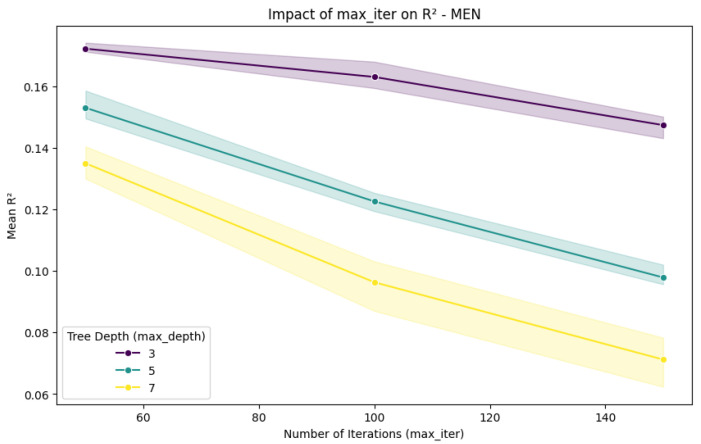
Impact of max_iter on $R^{2}$ in men.

**Figure 6 nutrients-17-01052-f006:**
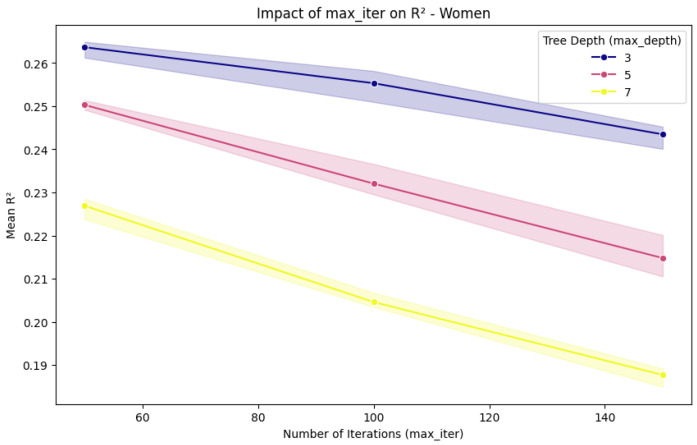
Impact of max_iter on $R^{2}$ in women.

**Table 1 nutrients-17-01052-t001:** Subset of clinical, anthropometric, and lifestyle features of women obtained through Boosted DTR, CatBoost, SHAP, and XGBoost.

Boosted DTR		CatBoost		SHAP		XGBoost	
**Feature**	**Importance**	**Feature**	**Importance**	**Feature**	**MASV**	**Feature**	**MASV**
Body Mass Index	0.0375	Triglycerides	7.2254	Creatinine	0.0138	Mother with diabetes	0.0073
Triglycerides	0.0214	Body Mass Index	6.8841	Body Mass Index	0.0125	Passive smoker	0.0045
Weight	0.0198	Creatinine	6.6062	Triglycerides	0.0109	BB4K	0.0044
HDL-cholesterol	0.0140	Atherogenic index	5.4440	Atherogenic index	0.0093	Snoring	0.0040
Glucose	0.0139	Urinary sodium	4.9646	Glucose	0.0088	Triglycerides	0.0038
Creatinine	0.0137	Glucose	4.5729	Urinary sodium	0.0077	Menstrual cycle	0.0036
Atherogenic index	0.0135	Age	4.3607	Weight	0.0065	Smoke	0.0033
MGMD	0.0126	Weight	3.3334	Age	0.0064	Father with gout	0.0031
Waist circumference	0.0120	Waist circumference	2.9811	Iron	0.0038	ISSSTE	0.0030
Iron	0.0102	Total cholesterol	1.7219	Waist circumference	0.0033	Father with hypertension	0.0028
Urinary sodium	0.0100	Respiratory rate	1.4676	Heart rate	0.0026	Father with diabetes	0.0028
Father with gout	0.0100	Heart rate	1.3924	Mother with diabetes	0.0024	Mother with dyslipidemia	0.0027
Pregnancies	0.0093	Serum sodium	1.3627	Pregnancies	0.0023	Very low physical activity	0.0025
BB4K	0.0091	Height	1.2105	Urine potassium	0.0022	Skilled worker	0.0024
Father with obesity	0.0091	Mother with diabetes	1.1261	Complications in childbirth	0.0021	Mother with hypertension	0.0023

*BB4K*: baby birth weight over 4 kg; *MGMD*: maternal grandmother diabetes; *ISSSTE*: Institute for Social Security and Services for State Workers.

**Table 2 nutrients-17-01052-t002:** Performance metrics for the subset of clinical, anthropometric, and lifestyle features for women identified by Boosted DTR, CatBoost, SHAP, and XGBoost.

Model	Parameters	Method	MSE	RMSE	$R^{2}$
CatBoost	depth: 6 iterations: 300, l2_leaf_reg: 5 learning_rate: 0.05	Importance	0.0063	0.0793	0.2315
DTR	learning_rate: 0.01, max_depth: 4, n_estimators: 300, subsample: 0.8	Importance	0.0087	0.0930	0.2768
SHAP	alpha: 1, colsample_bytree: 0.3, learning_rate: 0.1, max_depth: 5, n_estimators: 100	SHAP	0.0068	0.0824	0.1688
XGBoost	alpha: 1, colsample_bytree: 0.3, learning_rate: 0.1, max_depth: 5, n_estimators: 100	MeanAbsShapValue	0.0079	0.0890	0.3170

**Table 3 nutrients-17-01052-t003:** Subset of clinical, anthropometric, and lifestyle features of men obtained through Boosted DTR, CatBoost, SHAP, and XGBoost.

BOOSTED DTR		CATBOOST		SHAP		XGBoost	
**Feature**	**Importance**	**Feature**	**Importance**	**Feature**	**MASV**	**Feature**	**MASV**
Body Mass Index	0.0330	Creatinine	8.6576	Body Mass Index	0.0181	Father with diabetes	0.0103
Waist circumference	0.0193	Body Mass Index	5.9170	Creatinine	0.0169	Passive smoker	0.0097
Creatinine	0.0173	Atherogenic index	4.1839	Atherogenic index	0.0106	Father with gout	0.0077
Daily smoker	0.0159	Triglycerides	3.8335	Triglycerides	0.0102	Mother with diabetes	0.0066
Atherogenic index	0.0158	Urinary sodium	3.4797	Weight	0.0074	IMSS	0.0055
MGFHA	0.0136	Waist circumference	3.2791	Urinary sodium	0.0068	Whiskey	0.0049
Currently smokes	0.0132	Age	2.6792	Age	0.0063	High cholesterol	0.0048
Triglycerides	0.0131	HDL-Cholesterol	2.3282	Father with diabetes	0.0062	Cognac	0.0044
Drunk	0.0131	Urine potassium	2.1635	Iron	0.0052	Father with obesity	0.0038
Father with gout	0.0120	Weight	1.8986	Passive smoker	0.0049	Use of electric vehicle	0.0036
Father with diabetes	0.0117	Height	1.8661	Waist circumference	0.0047	Biparental family	0.0033
Passive smoker	0.0110	LDL-Cholesterol	1.8497	Urine potassium	0.0045	PGMD	0.0032
Whiskey	0.0109	Total cholesterol	1.7534	Height	0.0042	MGMD	0.0032
Prof	0.0106	Pulse pressure	1.7148	IMSS	0.0034	Drunk	0.0031

*MGFHA*: maternal grandfather heart attack; *Prof*: professional academic level; *IMSS*: Mexican Social Security Institute; *PGMD*: paternal grandmother with diabetes; *MGMD*: maternal grandmother with diabetes.

**Table 4 nutrients-17-01052-t004:** Performance metrics for the subset of clinical, anthropometric, and lifestyle features for men identified by Boosted DTR, CatBoost, SHAP, and XGBoost.

Model	Parameters	Method	MSE	RMSE	$R^{2}$
CatBoost	depth: 6, iterations: 500, l2_leaf_reg: 1, learning_rate: 0.05	Importance	0.0115	0.1074	0.1687
DTR	learning_rate: 0.01, max_depth: 4, n_estimators: 300, subsample: 0.8	Importance	0.0111	0.1054	0.1994
SHAP	learning_rate: 0.1, max_depth: 3, n_estimators: 100	SHAP	0.0111	0.1054	0.1987
XGBoost	alpha: 1, colsample_bytree: 0.3, learning_rate: 0.1, max_depth: 5, n_estimators: 100	MeanAbsShapValue	0.0137	0.1072	0.1996

**Table 5 nutrients-17-01052-t005:** Subset of features from a food frequency questionnaire for men obtained through Boosted DTR, CatBoost and SHAP.

Boosted DTR		CatBoost		SHAP	
**Feature**	**Importance**	**Feature**	**Importance**	**Feature**	**MASV**
A medium cola soda	0.0239	A medium cola soda	6.9284	A medium cola soda	0.0110
Corn	0.0135	A corn tortilla	2.2745	A banana	0.0030
Safflower oil	0.0134	A banana	2.1508	A cup of atole with milk	0.0025
A teaspoon of margarine	0.0130	A portion of pork	2.0625	A teaspoon of margarine	0.0025
Corn oil	0.0128	A teaspoon of chocolate powder	1.7368	A portion of pork	0.0023
A tamale	0.0127	A liver steak or chicken liver	1.4353	A piece of chorizo or sausage	0.0022
A teaspoon of honey, jam or caramel	0.0125	A cup of atole with milk	1.4117	Chicken egg	0.0022
A teaspoon of chocolate powder	0.0122	A bowl of oatmeal	1.3965	A piece of sweet bread	0.0020
A portion of pork	0.0121	A bowl of sardines in tomato sauce	1.2908	A cup of beans	0.0019
A banana	0.0118	A cup of coffee without sugar	1.2908	Half a cup of seafood	0.0018
A bowl of oatmeal	0.0117	Olive oil	1.2625	Corn	0.0017
A cup of atole with milk	0.0117	A slice of white bread	1.2612	A liver steak or chicken liver	0.0017
A piece of chorizo or sausage	0.0115	Chicken egg	1.2538	Safflower oil	0.0016
A portion of carnitas	0.0115	Wheat tortilla	1.2270	Pork skin	0.0016
A teaspoon of vegetable shortening	0.0115	A fresh apple	1.2131	a teaspoon of chocolate powder	0.0014

**Table 6 nutrients-17-01052-t006:** Performance metrics for the subset of food frequency questionnaire features for men.

Model	Parameters	Method	MSE	RMSE	$R^{2}$
CatBoost	depth: 10, iterations: 500, l2_leaf_reg: 3, learning_rate: 0.01	Importance	0.0129	0.1135	0.0714
Boosted_DTR	learning_rate: 0.05, max_depth: 3, n_estimators: 200, subsample: 0.8	Importance	0.0128	0.1133	0.0739
SHAP	alpha: 1, colsample_bytree: 0.7, learning_rate: 0.01, max_depth: 5, n_estimators: 200	SHAP	0.0129	0.1136	0.0702

**Table 7 nutrients-17-01052-t007:** Performance metrics for the subset of food frequency questionnaire features for women.

Model	Parameters	Method	MSE	RMSE	$R^{2}$
CatBoost	depth: 10, iterations: 300, l2_leaf_reg: 1, learning_rate: 0.05	Importance	0.0111	0.1052	0.0763
DTR	learning_rate: 0.05, max_depth: 5, n_estimators: 100, subsample: 0.8	Importance	0.0111	0.1051	0.0767
SHAP	alpha: 1, colsample_bytree: 0.3, learning_rate: 0.1, max_depth: 20, n_estimators: 200	SHAP	0.0083	0.0913	0.0187

**Table 8 nutrients-17-01052-t008:** Subset of nutritional and biochemical features for women and men obtained from a food frequency questionnaire via CatBoost and SHAP.

Women				Men			
**CatBoost**		**SHAP**		**CatBoost**		**SHAP**	
**Feature**	**Importance**	**Feature**	**MASV**	**Feature**	**Importance**	**Feature**	**MASV**
Glucose	3.0789	Glucose	0.0050	Fructose	2.4942	Fructose	0.0044
Carbohydrates	2.8389	Starch	0.0033	Maltose	2.3573	Cholesterol	0.0039
Alcohol	2.1420	Alcohol	0.0032	Glucose	2.3070	Glucose	0.0038
Fructose	1.9236	Fructose	0.0030	Carbohydrates	1.9796	Maltose	0.0037
Maltose	1.9102	Beta cryptoxanthin	0.0030	Lactose	1.9027	Carbohydrates	0.0036
Cholesterol	1.8989			Alcohol	1.8901	Maltose	0.0034
						Beta cryptoxanthin	0.0020

**Table 9 nutrients-17-01052-t009:** Performance metrics for the subset of nutritional and biochemical features for women and men.

Sex	Model	Parameters	Method	MSE	RMSE	$R^{2}$
Men	CatBoost	depth: 10, iterations: 300, l2_leaf_reg: 1, learning_rate: 0.01	Importance	0.0130	0.1139	0.0654
Men	SHAP	alpha: 1, colsample_bytree: 0.7, learning_rate: 0.01, max_depth: 10, n_estimators: 200	SHAP	0.0135	0.1160	0.1345
Women	CatBoost	depth: 10, iterations: 500, l2_leaf_reg: 1, learning_rate’: 0.01	Importance	0.0106	0.1030	0.1139
Women	SHAP	alpha: 1, colsample_bytree: 0.3, learning_rate: 0.01, max_depth: 10, n_estimators: 500	SHAP	0.0077	0.0880	0.0523

**Table 10 nutrients-17-01052-t010:** Performance metrics of cross-feature method for groups in men and women.

Group	Sex	Parameters	MSE	RMSE	$R^{2}$
	Women	colsample_bytree: 0.9, learning_rate: 0.05, max_depth: 5, n_estimators: 50, subsample: 0.8	0.0034	0.0582	0.5857
Clinical, anthropometric and lifestyle	Men	colsample_bytree: 0.8, learning_rate: 0.05, max_depth: 3, n_estimators: 50, subsample: 0.8	0.0093	0.0964	0.3302
	Women	colsample_bytree: 0.9, learning_rate: 0.05, max_depth: 3, n_estimators: 100, subsample: 1.0	0.0091	0.0951	0.2431
Food frequency questionnaire	Men	colsample_bytree: 0.8, learning_rate: 0.01, max_depth: 5, n_estimators: 150, subsample: 0.9	0.0099	0.0995	0.2865
	Women	colsample_bytree: 0.8, learning_rate: 0.05, max_depth: 7, n_estimators: 50, subsample: 0.9	0.0039	0.0627	0.519
Nutritional and biochemical	Men	colsample_bytree: 0.8, learning_rate: 0.05, max_depth: 7, n_estimators: 50, subsample: 0.9	0.0096	0.0979	0.3098

**Table 11 nutrients-17-01052-t011:** Subset of features from a food frequency questionnaire for women obtained through Boosted DTR, CatBoost and SHAP.

Boosted DTR		CatBoost		SHAP	
**Feature**	**Importance**	**Feature**	**Importance**	**Feature**	**MASV**
A medium cola soda	0.0292	A medium cola soda	6.0152	A drink (rum, brandy, or tequila)	0.0078
A teaspoon of butter	0.0170	A portion of carnitas	2.8352	A medium cola soda	0.0065
A drink (rum, brandy, or tequila)	0.0165	A glass of sugary flavored water	2.0415	A bowl with rice	0.0039
A teaspoon of hot sauce or chili	0.0150	A cup of coffee without sugar	1.9075	Safflower oil	0.0034
A portion of carnitas	0.0141	A teaspoon of margarine	1.8767	A glass of sugary flavored water	0.0033
Pork skin	0.0136	A glass of whole milk	1.7528	Wheat tortilla	0.0032
A glass of sugary flavored water	0.0135	A cup of yogurt or Bulgarian yogurt	1.7215	A corn	0.0032
A corn tortilla	0.0134	Soy Oil	1.6787	A portion of carnitas	0.0029
A taco with al pastor meat	0.0132	A portion of dried beans	1.6536	A beer	0.0029
Corn oil	0.0132	A taco with al pastor meat	1.6423	A teaspoon of honey, jam or caramel	0.0027
A bowl of pasta soup	0.0130	A corn tortilla	1.6204	A cup of yogurt or Bulgarian yogurt	0.0025
Half a cup of seafood	0.0129	Olive oil	1.6147	A orange	0.0025
Half a cup of peas	0.0128	A cup of beans	1.5657	A portion of grapes	0.0025
A bowl of oatmeal	0.0128	Half a cup of peas	1.4594	A slice of papaya	0.0025
A sapote	0.0126	A slice of papaya	1.3905	A bowl of cream of vegetable soup	0.0023

## Data Availability

The original contributions presented in the study are included in the article/Appendix A, further inquiries can be directed to the corresponding authors due to the ethical reasons giving the sensitive nature of some of the data.

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
