# Peer review of "Tlalpan 2020 Case Study: Enhancing Uric Acid Level Prediction with Machine Learning Regression and Cross-Feature Selection"

_nutrients, 2025, doi:10.3390/nu17061052_

Round 1
Reviewer 1 Report
Comments and Suggestions for Authors
The manuscript by Gutiérrez-Esparza et al. aims to evaluate the factors influencing blood uric acid levels in a prospective longitudinal study of risk factors for hypertension incidence in a Mexico City population, with a particular focus on gender-specific differences. Using machine-learning predictive models with four algorithms, the authors identified critical variables with significant impact on the blood analyte. The study provided interesting insight that could be used for cost-effective healthcare interventions.
Overall, the background information, methodology, data presentation, tables, and conclusions are excellent. Graphs are acceptable. References are up to date and adequate in number. Here, I will limit myself to a few remarks.
Major comments
Page 3, lines 81-82. In general, the mortality curve for blood uric acid concentrations between 200 and 300 µmol/L has a non-linear trend (PMID: 24534458). In that narrow range, mortality decreases as concentration increases. Furthermore, in elderly people, there seems to be a uric acid paradox (PMID: 33820873), namely, a protective effect, perhaps due to the antioxidant properties of the molecule. Given that the participants in the Tlalpan 2020 project at baseline were younger than 50 years of age (line 82), do the authors think that their findings may have been affected by such limitation?
Pages 4-8. The list of variables in Table 1 is very long. Don't the authors think it could be reported as supplementary material rather than in the text?
Table 1. Some of the variables reported in Table 1 are marked as dichotomous. Among these are respiratory rate, body mass index, cholesterol (Chol), and “good cholesterol” (CholGd). In this way, by collapsing the variable into just two values, didn’t information get partially lost?
Page 8, lines 120-125, and page 20, lines 329-334. Cross-feature selection is used to reduce the number of predictive variables. The authors rightly pointed out that in this way, overfitting is limited, but it would have been helpful to explain it more extensively. How was overfitting estimated, and what measures were taken to eliminate irrelevant or duplicate information from the training data? For instance, the top critical features highlighted by the algorithms are likely to be mutually intercorrelated (body mass index, triglycerides, etc.). They would have been probably excluded by conventional (non-AI) statistical models that assess the effect of individual predictors.
Page 11, lines 226-231. It appears that overall, the four algorithms showed significantly superior performance in women compared to men (R² comparison between Table 3 and Table 5). Do the authors have any explanation for this sex-related difference?
Minor comments
Page 2, lines 32-33. “reference range” is preferable.
Page 6. If possible, write “Movilidad y Transporte” in English.
Page 9, line 155. Python 3.8 was released in October 2019 and is currently considered out of date.
Page 25, ref. no. 3. Please correct “Rheumatology 2024;63:2411-2417” and similar mistakes in the reference list.
Author Response
Comments and Suggestions for Authors
The manuscript by Gutiérrez-Esparza et al. aims to evaluate the factors influencing blood uric acid levels in a prospective longitudinal study of risk factors for hypertension incidence in a Mexico City population, with a particular focus on gender-specific differences. Using machine-learning predictive models with four algorithms, the authors identified critical variables with significant impact on the blood analyte. The study provided interesting insight that could be used for cost-effective healthcare interventions.
Overall, the background information, methodology, data presentation, tables, and conclusions are excellent. Graphs are acceptable. References are up to date and adequate in number. Here, I will limit myself to a few remarks.
The authors are grateful to Reviewer 1 for the insightful comments made about our work. We will follow the advice and counsel given. In what follows we will present a point-by-point response to the review. To ease reading our comments appear in bold-type.
Major comments
Page 3, lines 81-82. In general, the mortality curve for blood uric acid concentrations between 200 and 300 µmol/L has a non-linear trend (PMID: 24534458). In that narrow range, mortality decreases as concentration increases. Furthermore, in elderly people, there seems to be a uric acid paradox (PMID: 33820873), namely, a protective effect, perhaps due to the antioxidant properties of the molecule. Given that the participants in the Tlalpan 2020 project at baseline were younger than 50 years of age (line 82), do the authors think that their findings may have been affected by such limitations?
Thank you for this relevant observation: We have added the following text in the limitations section of the discussion:
Since the baseline participants of the Tlalpan 2020 cohort were younger than 50 years of age, the potential protective effect of uric acid observed in elderly populations (the "uric acid paradox") might not be evident in this cohort. Additionally, the non-linear relationship between uric acid and mortality within the 200–300 µmol/L range suggests that small variations in uric acid levels could have different health implications depending on the age and metabolic context of the population. This age-related effect should be considered when interpreting the predictive modeling outcomes, as the relationships observed in younger adults may not extrapolate directly to older populations.
Pages 4-8. The list of variables in Table 1 is very long. Don't the authors think it could be reported as supplementary material rather than in the text?
Thank you for your comment. We have moved Table 1 to the supplementary material.
Table 1. Some of the variables reported in Table 1 are marked as dichotomous. Among these are respiratory rate, body mass index, cholesterol (Chol), and “good cholesterol” (CholGd). In this way, by collapsing the variable into just two values, didn’t information get partially lost?
Thank you for your comment. In the case of the variables Chol and CholGd, these come from the general section of the questionnaire, where participants simply indicate whether they have high cholesterol and its type. On the other hand, the Blood and Urine Test Results section presents cholesterol levels and their subtypes. Regarding body mass index, we decided to categorize it dichotomously to facilitate its interpretation within the context of predictive models, which allowed us to more easily identify general patterns. However, the results analysis showed that BMI, although dichotomous, remains a potential predictor in the predictive models for uric acid, highlighting its relevance in this study, even when not considered continuously
Page 8, lines 120-125, and page 20, lines 329-334. Cross-feature selection is used to reduce the number of predictive variables. The authors rightly pointed out that in this way, overfitting is limited, but it would have been helpful to explain it more extensively. How was overfitting estimated, and what measures were taken to eliminate irrelevant or duplicate information from the training data? For instance, the top critical features highlighted by the algorithms are likely to be mutually intercorrelated (body mass index, triglycerides, etc.). They would have been probably excluded by conventional (non-AI) statistical models that assess the effect of individual predictors.
Thank you for your comment. Overfitting was controlled through cross-validation and hyperparameter tuning. Feature selection was performed using multiple algorithms (Boosted DTR, CatBoost, SHAP, and XGBoost) to minimize irrelevant variables. While some selected features may be intercorrelated, models like XGBoost and CatBoost can effectively capture non-linear relationships and interactions between predictors. These clarifications have been incorporated into the manuscript.
Page 11, lines 226-231. It appears that overall, the four algorithms showed significantly superior performance in women compared to men (R² comparison between Table 3 and Table 5). Do the authors have any explanation for this sex-related difference?
Thank you for your insightful comment. The higher R² values observed in women compared to men could be explained by several factors. First, women constituted a smaller proportion of the cohort, which likely reduced the variability in their data, leading to more accurate predictions. Additionally, the predictors for women might not have carried as much weight, as some variables may not have been as representative or did not capture the full variability in the cohort. This could have resulted in more stable model performance, but with a lesser predictive impact. To further improve the algorithms, it would be crucial to continue collecting more data, which could enhance the representation of key variables and ultimately improve the models’ predictive power. We included the exact number of men and women in the cohort in the revised manuscript to further clarify this aspect and a note has been added in the discussion.
Minor comments
Page 2, lines 32-33. “reference range” is preferable.
We have corrected this term.
Page 6. If possible, write “Movilidad y Transporte” in English.
We have corrected this term.
Page 9, line 155. Python 3.8 was released in October 2019 and is currently considered out of date.
Thank you for your observation. We acknowledge that Python 3.8 is an older version. To ensure that the results remain consistent across different Python versions, we re-executed our analysis using Python 3.11. The key performance metrics and feature importance rankings remained stable. We have updated the manuscript to reflect the newer Python version.
Page 25, ref. no. 3. Please correct “Rheumatology 2024;63:2411-2417” and similar mistakes in the reference list.
We have corrected this.
Reviewer 2 Report
Comments and Suggestions for Authors
Tlalpan 2020 cohort study has already produced the papers in the series of data. The study was well done and written. Some modifications are potentially suggested for the increase of the paper’s quality.
- How much was the bias of cohort populations to generalize the results observed in the study? Even if it is speculation, we want to have more comments from the authors.
- Prior to the study, could the authors expect any difference in the results of respective models’ performance observed in the study across the models? The differences might be expected with prior references. If so, the hypothesis may be stated in Introduction.
- The authors stated the results would have public health implications. Although we agree with the statement, can the authors raise more concrete implications?
- Also, how about the use of the results in the clinical settings (i.e., cardiology clinics)?
- How much do the uric acid concentrations change day by day (or seasonally) within the individual? Is that change different between the individuals with and without the uric acid-related genetic predisposition? Can this change affect the results?
- Lines 23-25 in Abstract: the significance of gender-specific analyses was stated. We can look at the results of men. On the other hand, what was the results of women? Any comments may be included in the context.
- The expression ‘dietary and lifestyle’ was used in Line 26 in Abstract, while ‘lifestyle and nutritional’ was used in Line 483. The term ‘dietary’ or ‘nutritional’ can be unified throughout the manuscript.
- In theory, are such dietary/nutritional factors included in lifestyle factors?
- Line 199: a space might be removed before CatBoost.
- Some tables: the abbreviations would be fully spelled out in footnotes.
- Ref 13: ‘H’ypertension may be suitable as a Journal name.
- Refs14 and 16: we want to have URL or something we can access easily.
Author Response
Comments and Suggestions for Authors
Tlalpan 2020 cohort study has already produced the papers in the series of data. The study was well done and written. Some modifications are potentially suggested for the increase of the paper’s quality.
The authors want to acknowledge Reviewer 2 for the professional academic reviewing made about our work. We will follow the comments and suggestions given. In what follows, we will present a point-by-point response to the review. To ease reading our comments appear in bold-type.
How much was the bias of cohort populations to generalize the results observed in the study? Even if it is speculation, we want to have more comments from the authors.
The following text was added to the discussion:
The degree of bias in cohort populations when generalizing the results observed in the Tlalpan 2020 study depends on several factors, particularly the representativeness of the participants relative to the general population. One key limitation is the age restriction, as all participants were younger than 50 years at baseline, which may prevent the findings from being extrapolated to older populations where the relationship between uric acid and health outcomes differs, such as the observed uric acid paradox in the elderly. Additionally, geographical and socioeconomic factors play a role, as the cohort consists of residents of Mexico City, potentially limiting the applicability of the results to populations with different dietary habits, genetic backgrounds, and environmental exposures. Selection bias is another concern, as individuals who chose to participate may have distinct health statuses, lifestyles, or access to healthcare, making them non-representative of the broader population at risk of hyperuricemia-related conditions. To mitigate these biases, sensitivity analyses, external validation in diverse populations, and age-stratified modeling approaches may enhance the robustness and applicability of the findings.
Prior to the study, could the authors expect any difference in the results of respective models’ performance observed in the study across the models? The differences might be expected with prior references. If so, the hypothesis may be stated in the Introduction.
Thank you for your comment. While previous studies have reported variations in machine learning model performance for metabolic risk prediction, the objective of this study was primarily exploratory. Rather than testing a specific hypothesis about which model would perform best, we aimed to compare multiple algorithms under the same dataset conditions to identify the most relevant predictors of uric acid levels. Given the variability in model performance across different datasets and feature selections, we considered an empirical evaluation approach to be more appropriate. We have clarified this point in the Introduction of the revised manuscript.
We have revised the Introduction to acknowledge that model performance can vary across metabolic studies. We now clarify that this study took an exploratory approach to compare different machine learning algorithms and identify the most effective predictors of uric acid levels.
The authors stated the results would have public health implications. Although we agree with the statement, can the authors raise more concrete implications?
We have added the following text in the revised manuscript:
The public health implications of the study could be further strengthened by specifying how the predictive modeling of uric acid levels can inform targeted interventions and policy recommendations. For instance, if machine learning models accurately identify dietary and metabolic predictors of hyperuricemia, these insights could be used to develop personalized nutritional guidelines or community-based prevention programs aimed at reducing the burden of related conditions such as gout, kidney disease, and cardiometabolic disorders. Additionally, integrating these predictive tools into clinical practice could help healthcare professionals identify at-risk individuals earlier and implement lifestyle modifications before the onset of complications. Policymakers could also leverage these findings to design educational campaigns or refine dietary recommendations tailored to specific demographic groups. By explicitly outlining these applications, the authors could provide a clearer connection between their findings and actionable public health strategies.
Also, how about the use of the results in the clinical settings (i.e., cardiology clinics)?
To further address this issue, we have added the following text in the revised manuscript:
The results of this study could have valuable applications in clinical settings, particularly in cardiology clinics, where hyperuricemia is increasingly recognized as a potential risk factor for cardiovascular disease. If machine learning models can accurately predict uric acid levels based on metabolic and dietary factors, clinicians could integrate these tools into routine assessments to identify patients at higher risk of developing conditions such as hypertension, atherosclerosis, or heart failure. This predictive capability would allow for earlier interventions, including lifestyle modifications, dietary counseling, and, when necessary, pharmacological strategies to manage uric acid levels proactively. Moreover, incorporating these findings into risk stratification models could enhance personalized treatment approaches, ensuring that patients receive tailored recommendations based on their unique metabolic profiles. By bridging predictive analytics with clinical decision-making, this research has the potential to improve preventive care and optimize cardiovascular risk management.
How much do the uric acid concentrations change day by day (or seasonally) within the individual? Is that change different between the individuals with and without the uric acid-related genetic predisposition? Can this change affect the results?
We acknowledge that uric acid concentrations can fluctuate daily and seasonally due to various physiological and environmental factors. However, as our study is based on a cross-sectional analysis using baseline data, it does not capture these variations over time. While individuals with a genetic predisposition to hyperuricemia might experience different fluctuation patterns, this aspect was not assessed in our dataset. Future longitudinal analyses incorporating repeated measurements would be needed to explore these dynamics. We have clarified this limitation in the manuscript.
Lines 23-25 in Abstract: the significance of gender-specific analyses was stated. We can look at the results of men. On the other hand, what were the results of women? Any comments may be included in the context.
Thank you for your insightful comment. We have revised the Abstract to include key findings for women.
The expression ‘dietary and lifestyle’ was used in Line 26 in Abstract, while ‘lifestyle and nutritional’ was used in Line 483. The term ‘dietary’ or ‘nutritional’ can be unified throughout the manuscript.
Thank you for your comment. We have standardized the terminology throughout the manuscript using the more widely used term “nutritional”.
In theory, are such dietary/nutritional factors included in lifestyle factors?
Thank you for your comment. Although dietary/nutritional factors might conceptually fall under lifestyle, in our study they were assessed using a separate food frequency questionnaire. This approach allowed us to distinctly capture overall lifestyle behaviors (e.g., physical activity, smoking) and specific nutritional intake.
Line 199: a space might be removed before CatBoost.
We have corrected the spacing issue.
Some tables: the abbreviations would be fully spelled out in footnotes.
This issue has been corrected.
Ref 13: ‘H’ypertension may be suitable as a Journal name.
This issue has been corrected
Refs14 and 16: we want to have URLs or something we can access easily.
Thank you for your comment. Reference 14 corresponds to a printed book, and while there is no direct online access to the full text, additional information about the book and its standards can be found at https://www.isak.global/. The book is primarily available through libraries and institutions, as indicated in catalog records such as WorldCat. Similarly, Reference 16 is based on printed materials, including the questionnaire manual and the food frequency questionnaire. Since these materials are not available online, we can provide them as attached documents for your reference, whenever you request them.
Reviewer 3 Report
Comments and Suggestions for Authors
This study presents a well-structured application of machine learning methodologies for predicting uric acid levels, leveraging a diverse dataset that integrates clinical, anthropometric, and lifestyle variables. Its thorough documentation of model selection, feature engineering, and validation provides a valuable contribution to the growing field of predictive health analytics. However, several aspects merit further consideration:
- The study is based on the Tlalpan 2020 cohort from Mexico City, which may not be representative of other populations with distinct ethnic, socioeconomic, or geographic characteristics​. Would the inclusion of external validation cohorts from different regions enhance the applicability and robustness of the conclusions drawn?
- Lifestyle habits and dietary intake were obtained through self-reported questionnaires, which are subject to recall bias and underreporting​. Could the integration of objective dietary assessment methods, such as biochemical markers or digital food tracking, improve data reliability?
- The study provides a cross-sectional analysis of uric acid levels without examining long-term changes​. Would a longitudinal approach allow for a more nuanced understanding of the impact of dietary and lifestyle modifications over time?
- While the study accounts for gender and lifestyle factors, other potential confounders, such as genetic predisposition and concurrent medication use, are not extensively explored​. Could additional genetic analysis or pharmacological data strengthen the causal inferences made?
- The use of advanced machine learning models like XGBoost and CatBoost enhances predictive accuracy but may reduce interpretability​. How could explainability techniques, such as SHAP values, be further leveraged to provide clearer insights into the relationships between features and outcomes?
6.The Cross-Feature Selection method appears to improve predictive performance, but its impact on model stability is not fully assessed​. Would additional sensitivity analyses or comparison with alternative feature selection techniques provide more confidence in its effectiveness?
7.The study suggests that machine learning can inform public health strategies for hyperuricemia prevention​. Could the findings be translated into actionable dietary recommendations or screening protocols to optimize clinical utility?
Author Response
Comments and Suggestions for Authors
This study presents a well-structured application of machine learning methodologies for predicting uric acid levels, leveraging a diverse dataset that integrates clinical, anthropometric, and lifestyle variables. Its thorough documentation of model selection, feature engineering, and validation provides a valuable contribution to the growing field of predictive health analytics. However, several aspects merit further consideration:
The authors want to thank Reviewer 3 for the insightful reviewing made about our work. We will follow the comments and suggestions given. In what follows, we will present a point-by-point response to the review. To ease reading our comments appear in bold-type.
The study is based on the Tlalpan 2020 cohort from Mexico City, which may not be representative of other populations with distinct ethnic, socioeconomic, or geographic characteristics​. Would the inclusion of external validation cohorts from different regions enhance the applicability and robustness of the conclusions drawn?
Thank you for your comment. We acknowledge that the Tlalpan 2020 cohort, based in Mexico City, may not fully represent populations with different ethnic, socioeconomic, or geographic characteristics. While our findings provide valuable insights into this specific cohort, we recognize that including external validation cohorts from diverse regions would enhance the generalizability and robustness of the conclusions. However, such validation was beyond the scope of this study. Future research could benefit from multi-regional cohorts to assess the consistency of our findings across different populations. We have included this limitation at the end of the Discussion section
Lifestyle habits and dietary intake were obtained through self-reported questionnaires, which are subject to recall bias and underreporting​. Could the integration of objective dietary assessment methods, such as biochemical markers or digital food tracking, improve data reliability?
Thank you for your comment. We acknowledge that self-reported dietary data may be subject to recall bias and underreporting. Unlike biochemical markers, which reflect a single point in time, food frequency questionnaires allow for the analysis of habitual intake. While digital food tracking could improve accuracy, its implementation in large-scale studies presents logistical and financial challenges. To mitigate bias, participants received detailed instructions, and their responses were cross-checked with other lifestyle factors. We have added this limitation in the Limitations section.
The study provides a cross-sectional analysis of uric acid levels without examining long-term changes​. Would a longitudinal approach allow for a more nuanced understanding of the impact of dietary and lifestyle modifications over time?
Although the Tlalpan 2020 cohort is designed as a longitudinal study, this analysis focuses on baseline data, as it provides the most complete and consistent set of variables across participants. This cross-sectional approach allows for the exploration of initial associations between uric acid levels, dietary intake, and lifestyle factors before assessing long-term changes. We have clarified this in Section 2.1 Data of the manuscript. Future analyses will incorporate follow-up data to evaluate how these factors influence uric acid levels over time.
While the study accounts for gender and lifestyle factors, other potential confounders, such as genetic predisposition and concurrent medication use, are not extensively explored​. Could additional genetic analysis or pharmacological data strengthen the causal inferences made?
Thank you for your comment. Our dataset does not include genetic markers or detailed medication records, limiting our ability to assess their impact on uric acid levels. While these factors may of course strengthen causal inferences, they were beyond the scope of this study. This clarification has been added at the end of the Discussion section.
The use of advanced machine learning models like XGBoost and CatBoost enhances predictive accuracy but may reduce interpretability​. How could explainability techniques, such as SHAP values, be further leveraged to provide clearer insights into the relationships between features and outcomes?
Thank you for your insightful comment. To improve interpretability, we leveraged SHAP values to quantify feature importance and enhance model transparency. In our study, we applied SHAP to analyze relationships among key variables, explore sex-specific differences, and visualize distinct risk patterns through dependency plots and clustering-based SHAP analyses. These methods strengthened our interpretation of the results, and we have refined the discussion accordingly. This explanation has been added at the end of Section 4.3 ("Nutritional and Biochemical Factors").
6.The Cross-Feature Selection method appears to improve predictive performance, but its impact on model stability is not fully assessed​. Would additional sensitivity analyses or comparison with alternative feature selection techniques provide more confidence in its effectiveness?
To assess the stability of the Cross-Feature Selection method, we conducted a sensitivity analysis by varying max_iter, max_depth, and l2_regularization using HistGradientBoostingRegressor. The results indicate that predictive performance remains stable across different hyperparameter configurations. Figures and tables with detailed results have been added to the manuscript and supplementary material.
7.The study suggests that machine learning can inform public health strategies for hyperuricemia prevention​. Could the findings be translated into actionable dietary recommendations or screening protocols to optimize clinical utility?
Our study provides valuable insights into the relationship between uric acid levels, dietary intake, and lifestyle factors, contributing to the broader understanding of hyperuricemia risk. While the implementation of public health policies is beyond the scope of this study, our findings could inform future initiatives aimed at early detection and prevention. Additionally, efforts are being considered to develop predictive models that could enhance risk assessment and support data-driven decision-making. We have included this clarification in the Conclusion section of the revised manuscript.
Round 2
Reviewer 2 Report
Comments and Suggestions for Authors
The paper was much improved.
Comments on the Quality of English LanguageIt was good.